# Persistent Elevation in Urinary Neutrophil Gelatinase-Associated Lipocalin Levels Can Be a Predictor of Urinary Tract Infection Recurrence or Persistence in Women

**DOI:** 10.3390/ijms252312670

**Published:** 2024-11-26

**Authors:** Min-Ching Liu, Yuan-Hong Jiang, Jia-Fong Jhang, Tien-Lin Chang, Chia-Cheng Yang, Hann-Chorng Kuo

**Affiliations:** Department of Urology, Hualien Tzu Chi Hospital, Buddhist Tzu Chi Medical Foundation, and Tzu Chi University, Hualien 970, Taiwan; whiteseal1233@gmail.com (M.-C.L.); redeemerhd@gmail.com (Y.-H.J.); alur1984@hotmail.com (J.-F.J.); zxc13912@gmail.com (T.-L.C.); ycc39946@gmail.com (C.-C.Y.)

**Keywords:** urinary tract infection, recurrence, urine cytokines, women

## Abstract

Women commonly experience urinary tract infection (UTI) recurrence. However, there is no effective tool for predicting recurrent UTI after the first UTI episode. Hence, this study aimed to investigate potential urinary inflammatory biomarkers and specific biomarkers for predicting UTI recurrence or persistence after antibiotic treatment in women. Forty women who had a history of recurrent UTI within 1 year after the initial episode and acute bacterial cystitis were treated with broad-spectrum antibiotics for 1 week. To measure inflammatory biomarker levels, urine samples were collected at the baseline and after 1 week, 1 month, and 3 months. The levels of urinary pro-inflammatory proteins such as neutrophil gelatinase-associated lipocalin (NGAL), nerve growth factor, CXC-motif chemokine ligand (CXCL)-1, interleukin-8, CXCL-10, monocyte chemoattractant protein-1, and tumor necrosis factor-alpha were measured using commercial kits. Seven healthy age-matched women were included as controls. The changes in urinary biomarker levels at the baseline and various time points were compared between women with and without UTI recurrence within 1 month or within 3 months after the initial antibiotic therapy. At the baseline, patients with a higher urinary white blood cell count had a significantly higher NGAL level than the controls and those with a low white blood cell count. Of the 40 patients with a history of recurrent UTI, 12 presented with UTI persistence or recurrence within 1 month and 19 within 3 months after the initial antibiotic treatment. Among the 28 patients without UTI recurrence at 1 month after treatment, 7 had UTI recurrence within 3 months. Compared with patients without UTI recurrence, those with UTI recurrence had significantly higher urinary NGAL levels at 1 week, 1 month, and 3 months after the initial treatment. This study concludes that persistent elevation in urinary NGAL levels after the initial antibiotic treatment indicated persistent bladder inflammation. Further, it could be a predictor of UTI persistence or recurrence within 1 or 3 months after the initial antibiotic treatment. Patients with a history of recurrent UTI and high urinary NGAL levels after antibiotic treatment might require a longer treatment duration to completely eradicate or prevent UTI recurrence.

## 1. Introduction

Recurrent urinary tract infection (UTI) is defined as repeated UTIs with a frequency of at least two episodes in the preceding 6 months or three episodes within 1 year. Recurrent UTI is commonly observed in women [1]. This complex condition requires antibiotic therapy, and it has a significant impact on women of all ages [2]. Recurrent UTI has a bothersome effect on the quality of life of patients, encompassing bladder pain, urinary frequency and urgency, anxiety, and clinical depression symptoms, among women who are affected [3]. Antibiotic treatment is an effective front-line therapy for UTIs. However, UTI recurrence remains a common issue, and it may occasionally be extremely troublesome. One in three women will experience a clinically significant UTI by the age of 24 years, and almost half of them will have at least one UTI episode in their lifetime. Other studies showed that 53% of women aged >55 years presented with UTI recurrence [4,5].

Currently, UTI is diagnosed based on clinical symptoms and urinalysis and urine culture results, which can quantify the presence of bacteria and the white blood cell (WBC) count in the urine [6]. Antibiotic treatment is usually discontinued after the resolution of bladder symptoms, and there are no accurate urinary biomarkers that can predict the risk of UTI recurrence after the first UTI episode [7]. Chronic bladder wall inflammation is associated with increased apoptosis and decreased expression of the tight junction proteins of the urothelium [8]. Based on this notion, we hypothesized that chronic bladder wall inflammation may persist after an acute UTI episode, which can potentiate UTI recurrence [9]. Inflammatory responses are activated during a UTI episode, and innate immunity responses result in elevated urinary cytokine levels, which facilitate the rapid clearance of infectious pathogens [10]. Among the urinary inflammatory biomarkers, interleukin (IL)-8, prostaglandin, and IL-13 have diagnostic value, as they are more effective in detecting recurrent UTI [11]. Neutrophil gelatinase-associated lipocalin (NGAL) modulates neutrophil functions and is employed in the diagnosis of acute kidney injury and bacterially mediated inflammatory processes [12]. The urinary NGAL level is much higher in patients with symptomatic UTI and has been used to detect early UTI in children and adults [13]. This study aimed to investigate potential urinary inflammatory biomarkers and specific urinary inflammatory biomarkers for predicting persistence or recurrence of UTI after the initial antibiotic treatment in women with a history of recurrent UTI, which means that continued antibiotics treatment may be necessary for a complete eradication of a UTI.

## 2. Results

This study initially enrolled 42 women with a history of recurrent UTI (UTI group) and 7 women without a history of recurrent UTI (control group). The mean ages of the UTI and control groups were 64.1 ± 14.9 and 57.4 ± 21.0 years, respectively (*p* = 0.301). The symptoms of two patients were resolved, and these patients declined follow-up after the initial antibiotic treatment. The other 40 women were followed for up to 3 months. Three patients had UTI persistence for up to 1 week, which was resolved after continuing antibiotics treatment. Within 1 month after the initial treatment, 12 (30%) of the 40 patients had UTI persistence (n = 9) or recurrence (n = 3), and 7 (25%) of the 28 patients who did not exhibit UTI persistence or recurrence within 1 month after the initial treatment had UTI recurrence within 1–3 months. Compared with the controls, patients with UTI persistence or recurrence had significantly higher overactive bladder symptoms at the baseline (7.96 ± 3.82 vs. 4.43 ± 4.08, *p* = 0.028). Moreover, their symptom score decreased with time, and it did not differ between the controls and patients with a history of recurrent UTI.

The results of the baseline urine culture are shown in Table 1. There was no significant difference in the distribution of bacterial species between the 19 patients who had UTI persistence or recurrence and the 21 without UTI recurrence. Only one patient’s bacteria growth changed from *Klebsiella pneumoniae* to *E. coli.* The other patients showed the same bacterial growth in the repeat urine culture within 3 months.

At the baseline, patients with a history of recurrent UTI had higher urinary levels of NGAL, CXC-motif chemokine ligand (CXCL)-1, IL-8, CXCL-10, monocyte chemoattractant protein (MCP-1), and tumor necrosis factor-alpha (TNF-α) than the controls; however, only the NGAL level was significantly higher in patients with a history of recurrent UTI (Table 2). Patients with recurrent UTI were categorized into two subgroups (those with a low baseline WBC count [≤10/high power field [HPF]] and those with a high WBC count [>10/HPF]). The results showed that the high WBC count group had significantly higher urinary NGAL levels than the low baseline WBC count group. The patients with recurrent UTI were further categorized into three subgroups (patients with a baseline WBC count of ≤10/HPF, 10–75/HPF, and ≥75/HPF). The results revealed that only the NGAL level significantly differed among the control group (32.2 ± 46.0 ng/mL), WBC ≤ 10/HPF group (38.7 ± 39.3 ng/mL), WBC 10–75/HPF group (305 ± 472 ng/mL), and WBC ≥ 75/HPF group (670 ± 540 ng/mL) (*p* < 0.001). The other cytokines did not significantly differ among patients with different urinary WBC counts (Appendix A).

Figure 1 shows the changes in urine biomarker levels from the baseline to different time points (1 week, 1 month, and 3 months) in patients with recurrent UTI and a comparison with the baseline values of the controls. Patients with recurrent UTI had persistently elevated cytokine levels at 1 week and 1 month and decreased levels at 3 months, particularly in NGAL, NGF, IL-8, and TNF-α, after the initial antibiotic treatment. The other urine cytokines, such as CXCL-1, CXCL-10, and MCP-1, were also higher than those in the controls at the baseline but did not reach a significant difference, and the urine levels did not decrease from the baseline to 3 months (Appendix A).

In total, 12 (30%) of the 40 patients with a history of recurrent UTI who were followed up had UTI persistence (n = 9) or recurrence (n = 3) within 1 month after the initial antibiotic treatment, and another 7 patients had UTI recurrence within 1 to 3 months. Figure 2 shows a comparison of the urinary biomarker levels at the baseline and different time points between patients with and without UTI persistence or recurrence within 1 month after the initial treatment. A significantly higher level of urine cytokine at the baseline was only noted for NGAL between patients with UTI persistence or recurrence within one month (685 ± 657 vs. 193 ± 169 ng/mL, *p* = 0.026). The other urine cytokine levels did not show significant differences between subgroups (Appendix A). Patients without UTI persistence or recurrence within 1 month exhibited a significant reduction in NGAL, CXCL-1, IL-8, and TNF-α levels from the baseline through to 3 months after the initial treatment. However, the differences between the UTI subgroups were not significant, except for NGAL, which is the most remarkable urinary biomarker. The NGAL levels were significantly higher in patients with UTI persistence or recurrence than those without at the baseline and all time points, and the urinary levels of patients with UTI persistence or recurrence even increased at 1 week (803 ± 895 vs. 51.0 ± 74.5 ng/mL, *p* = 0.019), 1 month (824 ± 1003 vs. 28.1 ± 30.8 ng/mL, *p* = 0.019), and 3 months (527 ± 724 vs. 30.1 ± 38.3 ng/mL, *p* = 0.058) compared with those in patients without UTI persistence or recurrence.

Table 3 shows a comparison of the levels of urinary biomarkers at the baseline and various time points after the initial antibiotic treatment between the patients with (n = 19, 47.5%) and those without (n = 21) UTI persistence or recurrence within 3 months after the initial treatment. Patients with UTI persistence or recurrence had significantly higher NGAL levels at the baseline than those without, and the NGAL levels of patients with UTI persistence or recurrence were persistently elevated at 1 week, 1 month, and 3 months compared with that of patients without. At 3 months after the initial treatment, patients with UTI persistence or recurrence had higher levels of CXCL-1, IL-8, and TNF-α than those without.

Among the 40 patients with a history of recurrent UTI, 9 had persistent UTI and 10 had recurrent UTI after the latest UTI episode. At the baseline, the urinary NGAL level was highest in patients with persistent UTI (641 ± 640 ng/mL), second highest in patients with recurrent UTI (300 ± 389 ng/mL), and lowest in patients without UTI recurrence (182 ± 168 ng/mL). The urinary NGAL level did not decrease with time in patients with persistent UTI, but it decreased in the recurrent UTI and no recurrent UTI subgroups. Patients with persistent UTI also had a higher urinary CXCL-1 level at 1 week and persistently elevated levels at 1 month and 3 months. Patients without UTI recurrence had significantly decreased urinary levels of NGAL, CXCL-1, IL-8, and CXCL-10 at 1 week, 1 month, and 3 months after antibiotics treatment (Figure 3, Appendix A).

## 3. Discussion

Our study showed that high levels of urinary inflammatory biomarkers such as NGAL, CXCL-1, IL-8, CXCL-10, MCP-1, and TNF-α are detected during acute UTI episodes in patients with a history of recurrent UTI. Persistent elevation in the level of urinary NGAL after the initial antibiotic treatment could predict UTI persistence or recurrence within 1 or 3 months in these women. The most important finding of this study is that a simple noninvasive diagnostic test could be useful when evaluating women who are at risk of UTI recurrence, and a longer duration of antibiotic treatment for UTI might be necessary.

Women who have a history of recurrent UTI, mainly due to an abnormal lower urinary tract function, including low bladder capacity, high voiding pressure, and detrusor overactivity, usually present with recurrence of bacterial cystitis [14]. Antibiotic treatment can usually eradicate acute UTI. However, a high percentage of women with recurrent UTI might develop a bacterial infection due to a resistance to the prophylactic antibiotics [15]. The infected bladder might continue to stay in a chronic inflammatory state, resulting in defective urothelial cell proliferation and barrier function in women with recurrent UTI [16]. Therefore, long-term antibiotic treatment or prophylaxis might be required to avoid upper urinary tract infection or renal function impairment [15]. The main pathophysiology of the defective bladder barrier function might be caused by inadequate bacterial clearance and prolonged inflammation [16]. If a noninvasive urine biomarker for detecting unresolved bladder inflammation after an acute UTI episode is available, appropriate antibiotic treatment could be continued until the biomarker levels decrease to an acceptable level, and UTI recurrence within the next few months or years might be prevented [17,18].

In this study, 30% of patients experienced UTI persistence or recurrence within 1 month after the initial antibiotic treatment. Moreover, 47.5% of women who had UTI persistence or recurrence within 3 months after the initial antibiotic treatment had a significantly higher NGAL level at the baseline, and the urinary NGAL level did not decrease at 1 week, 1 month, and up to 3 months. This finding indicated that the bladder inflammatory condition persisted in the patients who were at risk of UTI recurrence. The urinary NGAL level can be used to predict the presence of bladder inflammation and tissue damage in patients with UTI recurrence. Moreover, it may enhance bacterial colonization within the bladder urothelium and cause a breakthrough UTI when innate immunity is inadequate [19]. NGAL is present in neutrophils and other tissues and is essential in the innate antimicrobial immune system [12]. Elevated NGAL levels were detected in patients with bacterial UTI and acute kidney injury [20]. The urinary NGAL level was associated with the urinary WBC count but was weakly correlated with the bacterial count [21]. Based on previous studies, the urinary NGAL level can be used to diagnose UTI and acute pyelonephritis, particularly in children and infants, and prompt antibiotic treatment to prevent vesicoureteral reflux and renal scarring [22,23,24,25].

In the current study, women with UTI recurrence and a urinary WBC count >10/HPF presented with significantly higher levels of urinary inflammatory cytokines, including NGAL, NGF, IL-8, and TNF-α. The patients with an acute UTI who had been treated with antibiotics and a urinary WBC count ≤10/HPFP did not have elevated urinary cytokine levels. Based on these findings, the levels of these urine cytokines will decrease after the initial antibiotic treatment, and the urinary cytokines are sensitive to bacterial infection. Urinary cytokine levels have been used to differentiate UTI symptoms from bladder hypersensitivity symptoms [26]. The other cytokines were also associated with UTI and might potentially be a noninvasive tool for predicting UTI recurrence and treatment strategy [27,28,29,30]. A previous cross-sectional survey evaluated different urinary cytokines in asymptomatic women with a history of recurrent UTI. The results revealed that patients with bacteriuria had significantly higher urinary IL-8 levels. Urinary IL-8 levels might have a predictive value in women with recurrent UTI [11]. In this study, the IL-8 levels of women with UTI recurrence within 1 or 3 months after the initial treatment were higher at the baseline and at 1 week, 1 month, and 3 months. However, due to the small number of patients and wide standard deviation, the differences in urinary IL-8 between patients with and without UTI recurrence were not significant.

Further, this study showed that patients with a history of recurrent UTI had a significantly higher Overactive Bladder Symptom Score at the baseline than the controls. Patients with recurrent UTI present with increased urothelial cell apoptosis, barrier function deficit, and chronic suburothelial inflammation. Thus, there is a close association between recurrent UTI and bladder inflammation [8]. In a bladder with recurrent UTI, NGF driving sensory nerve sprouting is associated with mast cell activation and bladder pain expression [31]. These findings are in accordance with those of a previous study that found elevated urinary inflammatory biomarker levels in patients with OAB. Therefore, OAB symptoms are associated with chronic bladder inflammation and might indicate an incomplete resolution of urothelial dysfunction in patients with recurrent UTI [32]. Therefore, there is a close association between OAB and recurrent UTI, and this association might be interconnected with a common pathophysiology.

A UTI occurring within 3 months might be persistent or recurrent. Some patients had a relapse of UTI symptoms after their initial antibiotics treatment, while others might have persistent pyuria despite antibiotics according to their urine culture results. This study aimed to investigate potential urinary inflammatory biomarkers and specific urinary inflammatory biomarkers for predicting persistence or recurrence of UTI after the initial antibiotic treatment. Patients with frequently recurring UTI might have chronic bladder inflammation, resulting in urothelial dysfunction and a barrier deficit [16]. In this study, patients with persistent or recurrent UTI had significantly higher baseline urine NGAL levels than those without UTI recurrence, and these did not decrease with time after antibiotics treatment. The patients with recurrent UTI also had higher NGAL levels than those without UTI recurrence at the baseline, but these would decrease after antibiotics treatment. Patients without UTI recurrence exhibited significant decreases in their urinary levels of NGAL, CXCL-1, IL-8, and CXCL-10 at 1 week, 1 month, and 3 months after antibiotics treatment. This result indicates the presence of inflammatory processes within the bladder wall in patients with persistent or recurrent UTI, which might enable the breakthrough infection of the newly invaded bacteria or intracellular bacteria community. Urinary biomarkers may be used not only for predicting UTI recurrence but also for monitoring the inflammatory condition of a patient’s bladder [11,14]. Monitoring urine cytokine levels, especially the NGAL, might help us understand the bladder inflammation status and provide low-dose antimicrobial prophylaxis for a longer period to adequately eradicate intracellular bacterial communities, reduce the bladder inflammation, and improve the urothelial barrier function.

The current study has several limitations. First, we did not calculate the sample size for the study, and the sample size was small, which might be the reason why the standard deviations of some urine cytokine levels such as CXCL-1, IL-8, and CXCL-10 are very large. The large standard deviations are likely caused by one or more patients with a significantly higher grade of bladder inflammation. This also reflects the fact that the case number of this preliminary study was small and caused insignificant differences in urinary cytokine levels between the UTI subgroups and controls. However, with the small patient number, the results of this study still provide a significant clinical implication of the important role of urinary NGAL levels to predict the persistence or recurrence of UTI in patients with a history of recurrent UTI. Some women with UTI recurrence took antibiotics before visiting our clinic, resulting in improved symptoms and reduced urine cytokine levels at the baseline. Moreover, these factors can reduce the strength of this study. In addition, the inflammatory cytokine levels were relatively unstable and might have intra-individual variations, which resulted in statistically insignificant comparisons between subgroups. Further, systemic inflammatory diseases and comorbidities and lower urinary tract conditions might affect the expressions of these urinary proteins.

## 4. Materials and Methods

This is an observational study searching for potential urinary inflammatory biomarkers and specific urinary inflammatory biomarkers for predicting persistence or recurrence of UTI after the initial antibiotic treatment in women with a history of recurrent UTI. The study prospectively and consecutively enrolled 40 women with videourodynamic results proving uncomplicated, symptomatic, and urine culture-proven bacterial cystitis who had made consecutive visits at the outpatient urology clinic of a single medical center (Hualien Tzu Chi Hospital, Buddhist Tzu Chi Medical Foundation, Hualien, Taiwan) from February 2023 to March 2024. All patients had a previous history of two UTIs within the last 6 months or three episodes within 1 year and had an acute UTI episode prior to or occurring during this outpatient clinic visit. However, none of them had febrile pyelonephritis or demonstrable functional or anatomical abnormalities of the upper urinary tract. Because all women had a history of recurrent UTI, some patients had self-treated with antibiotics according to their previous urine culture results and sensitivity test before the visit; however, the bladder symptoms persisted. Clinical assessment, urinalysis, and urine culture were performed, and broad-spectrum antibiotics such as flouroquinolones (ciproxacin, ofloxacin) or cephalosporines (cefmetazole) were prescribed. This study was approved by the Institutional Review Board and Ethics Committee of the Hualien Tzu Chi Hospital, Buddhist Tzu Chi Medical Foundation (approval no.: 111-102-B). Women with bladder irritation symptoms but negative urinalysis results and without UTI symptoms were also invited to provide mid-stream urine samples, which were used as control samples. All patients in the current study were informed about the study rationale and procedures, and a written informed consent form was obtained before enrollment.

Recurrent UTI was defined as at least three episodes of symptomatic and medically diagnosed UTIs in the last 12 months or two UTI episodes within 6 months prior to the time of patient enrollment. Recurrence of UTI was diagnosed if the voiding symptom recurred and urinalysis showed pyuria (WBC count > 10/ HPF) after a period of previous UTI episodes. Women who had acute bacterial cystitis were treated with broad-spectrum antibiotics for at least 1 week, and antibiotic therapy was discontinued if the patient’s urinalysis results were negative and the patient was symptom-free. Patients who still had pyuria and bladder symptoms were treated with antibiotics for 1 more week. Patients who had UTI persistence or recurrence within 1 month were continuously treated according to the previous culture results without repeat urine culture. However, for patients who had UTI recurrence in 1 to 3 months, urine culture was routinely performed.

To measure urinary inflammatory biomarkers, urine samples were collected from the controls and patients with UTI at the baseline (the first day of urological visit) and at 1 week, 1 month, and 3 months. Urine was self-voided by patients after feeling a full bladder sensation. Then, urinalysis, including urinary WBC count, and urine culture were performed simultaneously to confirm the presence of infection. Some patients who received antibiotics at home but still had bladder symptoms such as pain on micturition, urinary urgency and frequency, and lower abdominal discomfort were considered to still have UTI even though their urine WBC count was <10/HPF.

For testing, 50 mL urine samples were immediately placed on ice and transferred to the laboratory for preparation. The samples were centrifuged at 1800 rpm for 10 min at 4 °C. The supernatant was separated into aliquots in 1.5 mL tubes (1 mL per tube) and was preserved in a freezer at −80 °C. Before performing further analyses, the frozen urine samples were centrifuged at 12,000 rpm for 15 min at 4 °C, and the supernatants were used for subsequent evaluations.

The urinary pro-inflammatory protein levels, including NGAL, IL-8, CXCL-1, TNF-α, MCP-1, CXCL-10, and nerve growth factor (NGF), were measured using commercial kits. Inflammatory biomarkers, including cytokines and neurotrophins, in the urine samples were assayed using a commercial microsphere kit and the Milliplex^®^ human cytokine/chemokine magnetic bead-based panel kit (Millipore, Darmstadt, Germany). Seven targeted analytes were used for the multiplex kit. The following inflammatory cytokines were included: CXCL-1, IL-8, MCP-1, CXCL-10, and TNF-α (catalog number: HCYTA-60 K); NGF (catalog number: HADK2MAG-61 K); and NGAL (catalog number: HKI2MAG-99K). The laboratory procedures used to quantify these targeted analytes in this study were similar to those reported in our previous studies [17]. Seven healthy age-matched women were used as controls.

The changes in urinary biomarker levels after the UTI episode and at different time points after antibiotics treatment were compared between women with UTI and those without, between women with UTI recurrence and those without, and between women with UTI recurrence within 1 month after the initial antibiotics therapy and those with UTI recurrence within 3 months. The changes in urinary biomarkers among patients with persistent UTI, recurrent UTI, and no UTI recurrence were also compared for their declination with time. Continuous variables were presented as means ± standard deviations and categorical variables as numbers and percentages. For a clear demonstration of the variation in urinary cytokine levels from the baseline to different time points, the variables were presented as mean ± standard errors in the figures. The mean differences in clinical data and the levels of urine biomarkers among the various groups were analyzed using one-way analysis of variance, and a post hoc test was performed using Bonferroni’s correction. All calculations were performed using the Statistical Package for the Social Sciences software version 10.0 (IBM Inc., Chicago, IL, USA). A *p*-value of <0.05 was considered statistically significant. The STROBE was used for a checklist of the study and manuscript preparation (Appendix A).

## 5. Conclusions

Persistent elevation in the urinary NGAL level from the baseline to 1 month after the initial antibiotic treatment might be useful in predicting UTI persistence or recurrence within 3 months after the initial treatment. Patients with persistent elevation of urinary NGAL level after initial antibiotics treatment were associated with UTI persistence or recurrence within 3 months and should be continuously treated with antibiotics to eradicate UTI and reduce bladder inflammation.

## Figures and Tables

**Figure 1 ijms-25-12670-f001:**
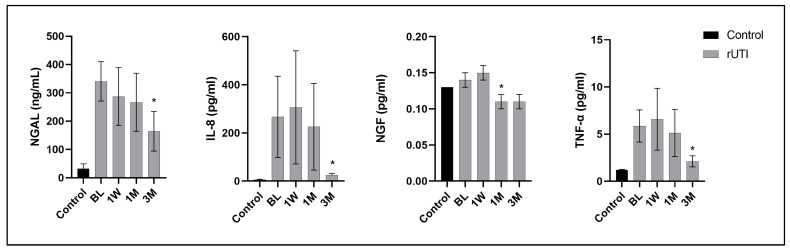
The changes in urine levels of NGAL, NGF, IL-8, and TNF-α from the baseline to different time points (1 week, 1 month, and 3 months) in the 40 patients with a history of recurrent urinary tract infection and comparison with the baseline values in controls. * indicates a significant difference with the baseline data. Data are expressed as mean ± standard error.

**Figure 2 ijms-25-12670-f002:**
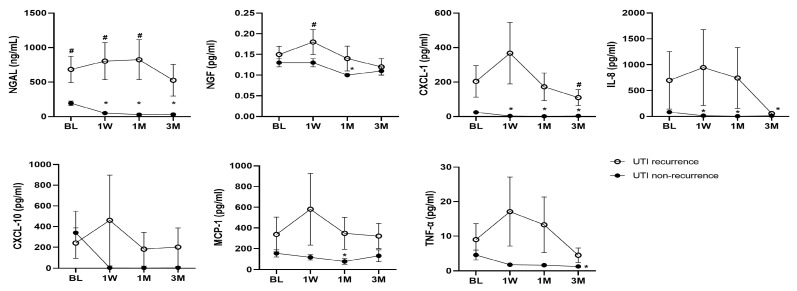
The urinary biomarker levels at the baseline and different time points of 12 patients with and 28 without UTI persistence or recurrence within 1 month after the initial treatment. * indicates a significant difference with the baseline data. # indicates a significant difference between groups. Data are expressed as mean ± standard error.

**Figure 3 ijms-25-12670-f003:**
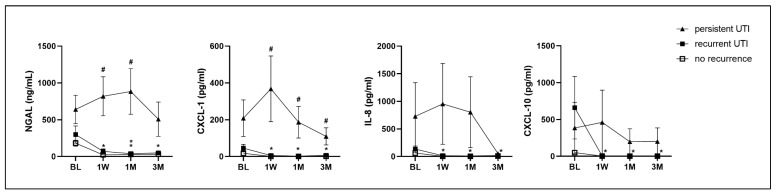
The changes in urinary cytokines NGAL, CXCL-1, IL-8, and CXCL-10 among patients with a history of recurrent UTI, who had persistent UTI (n = 9), recurrent UTI (n = 10), and no UTI recurrence (n = 21) within 3 months after the first UTI episode and antibiotics treatment. * significant change in urinary biomarker at a time point compared with the baseline in the same subgroup. # significant difference among the three subgroups at each time point.

**Table 1 ijms-25-12670-t001:** Baseline urine culture results for patients with and without persistence or recurrence of urinary tract infection (UTI).

	UTI Recurrence or Persistence (n = 19)	No UTI Recurrence or Persistence (n = 21)
*E. coli*	13	13
*Klebsiella pneunoniae*	1	2
*Enterococcus*	0	2
*Beta-streptococcus*	1	1
*Citrobacter*	1	0
No growth or normal flora	2	3
*Klebsiella aerogenes*	1	0

**Table 2 ijms-25-12670-t002:** Baseline urinary cytokine levels of controls and patients with a history of recurrent urinary tract infection and different baseline urinary leukocyte counts.

	Control (n = 7)	Total UTI (n = 40)	WBC ≤ 10/HPF (n = 8)	WBC > 10/HPF (n = 32)	*p*-Value *
NGAL	32.2 ± 46.0	340 ± 440	46.9 ± 45.34	414 ± 464	<0.001
NGF	0.13 ± 0.01	0.14 ± 0.04	0.13 ± 0.01	0.14 ± 0.05	0.590
CXCL-1	0.95 ± 0.7	78.1 ± 190	0.63 ± 0.23	97.4 ± 208	0.222
IL-8	5.21 ± 6.57	267 ± 1069	5.06 ± 5.17	332 ± 1190	0.583
CXCL-10	12.3 ± 23	310 ± 960	2.84 ± 3.28	387 ± 1062	0.403
MCP-1	149 ± 210	209 ± 353	41.5 ± 38.8	251 ± 384	0.262
TNF-α	1.22 ± 0.14	5.86 ± 10.8	1.16 ± 0.14	7.04 ± 11.9	0.182

* comparison between patients with a history of recurrent urinary tract infection with baseline WBC ≤ 10/HPF and WBC > 10/HPF and controls. NGAL: neutrophil gelatinase-associated lipocalin; NGF: nerve growth factor; CXCL-1: CXC-motif chemokine ligand-1; IL-8: interleukin-8; MCP-1: monocyte chemoattractant protein-1; TNF-α: tumor necrosis factor-alpha; WBC: white blood cell count per high power field; NGAL unit: ng/mL; cytokine unit: pg/mL.

**Table 3 ijms-25-12670-t003:** Urinary cytokine levels at baseline and various time points in patients with UTI persistence or recurrence within 3 months after initial antibiotic treatment.

	Time	Recurrent or Persistent UTI (n = 19)	No Recurrent or Persistent UTI (n = 21)	*p*-Value #
NGAL	Baseline	514 ± 567	183 ± 183	0.024
	1 week	524 ± 778	36.4 ± 35.2 *	0.017
	1 month	531 ± 878	27.8 ± 32.0 *	0.022
	3 month	354 ± 607	19.7 ± 18.1 *	0.044
NGF	Baseline	0.15 ± 0.05	0.13 ± 0.03	0.211
	1 week	0.16 ± 0.08	0.13 ± 0.04	0.186
	1 month	0.13 ± 0.08	0.10 ± 0.03 *	0.162
	3 month	0.12 ± 0.05	0.11 ± 0.05	0.344
CXCL-1	Baseline	141 ± 262	21.2 ± 36.1	0.064
	1 week	227 ± 489	1.76 ± 2.05 *	0.067
	1 month	109 ± 232	0.90 ± 0.83 *	0.057
	3 month	72.6 ± 125	1.32 ± 1.73 *	0.037
IL-8	Baseline	473 ± 1541	80.3 ± 122	0.252
	1 week	581 ± 1924	15.4 ± 16.7 *	0.235
	1 month	471 ± 1639	3.69 ± 3.53 *	0.199
	3 month	42.8 ± 57.3	10.6 ± 17.4 *	0.044
CXCL-10	Baseline	171 ± 409	436 ± 1269	0.391
	1 week	284 ± 1137	3.25 ± 3.58	0.317
	1 month	117 ± 446	2.08 ± 1.61	0.245
	3 month	130 ± 462	2.24 ± 4.14	0.211
MCP-1	Baseline	291 ± 480	135 ± 154	0.164
	1 week	386 ± 918	132 ± 123	0.266
	1 month	272 ± 447	56.4 ± 113	0.054
	3 month	314 ± 468	83.5 ± 99.1	0.071
TNF-α	Baseline	6.5 ± 13.1	5.29 ± 8.64	0.728
	1 week	11.1 ± 26.6	1.82 ± 0.91	0.162
	1 month	9.06 ± 22.6	1.58 ± 0.42	0.136
	3 month	3.35 ± 5.31	1.18 ± 0.48 *	0.125

* *p* < 0.05 compared with the baseline in the same patient group; # *p*-values between recurrent UTI and no recurrent UTI subgroups. Patients with recurrent or persistent UTI included 12 who had UTI recurrence or persistence within 1 month, and 7 within 1 to 3 months. Abbreviations: same as in the footnotes of Table 2.

## Data Availability

The data presented in this study are available on request from the corresponding author. The data are not publicly available due to regulation of the Institutional Review Board of the hospital.

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
