# Peer review of "Persistent Elevation in Urinary Neutrophil Gelatinase-Associated Lipocalin Levels Can Be a Predictor of Urinary Tract Infection Recurrence or Persistence in Women"

_ijms, 2024, doi:10.3390/ijms252312670_

Round 1
Reviewer 1 Report
Comments and Suggestions for Authors
Review of ijms-3277863
This manuscript describes the urinary levels of several pro-inflammatory proteins in control patients and those with a history of recurrent urinary tract infections. The goal was to determine if any of these proteins could be used as a predictive biomarker for the diagnosis of recurrent UTIs in other patients. The manuscript is one of a long series of papers from this research group, which have been published in various journals and cited in PubMed. The work appears to have been carefully done and the results are potentially very important. However, there are some missing experimental details and the presentation of the results is awkward. I think that the manuscript could be accepted for publication in the International Journal of Molecular Sciences after major revisions.
The following are more specific comments and suggestions.
1. The participants are described as 40 women with a history of recurrent uncomplicated urinary tract infections and 7 age-matched controls. What was the cause of the UTI in those patients that had one? While it might have been Escherichia coli, there are other bacteria that can cause similar infections. The authors say that broad spectrum antibiotics were prescribed as part of an earlier self treatment or the initial clinical assessment, but which ones were used is not given. More extensive microbiological and clinical history would be helpful. As the authors note in the Discussion, this study utilized a small sample size, particularly for the controls. Why were so few patients without symptomatic UTIs used?
2. The laboratory procedures are described very briefly and only include a listing of the commercial kits used. Reference 14 is given as a source of the methods for analysis, but this reference is actually a review article. I looked online and found an earlier paper from this group which gives these methods in more detail (Jiang, YH et al., 2020, Am. J. Physiol. Renal Physiol. 318: F1391-1399). It would be better either to summarize the methods as was done in this paper or to cite it as a reference. Protein concentrations are given in ng/ml in the legend to Table 1, but pg/ml in the earlier paper by Jiang et al. The authors should include the correct units in the Materials and Methods section.
3. Of the 40 patients with a history of recurrent UTI, 12 had a recurrence in one month and 7 more had a recurrence in one to three months. The total percentage of recurrence thus appears to have been 19/40 or 47.5%. Again, this indicates a small sample size for the analysis. In the case of those patients who showed a UTI recurrence, was there any microbiological analysis? Was the infection due to the same bacterial pathogen as the initial infection or a different one?
4. Table 1 shows the baseline urinary cytokine levels in the control group (n = 7) and in the recurrent UTI group, subdivided into those with a urinary white blood cell (WBC) count of <10 (n = 10) and those with a urinary white blood cell (WBC) count of >10 (n = 30). The p value refers to the difference between the two white blood cell groups. The values for several of the cytokines were significantly higher in the high WBC group than in the low WBC group. Did the authors assign a value to the severity of the clinical symptoms in these patients? Did this correlate with the WBC count? I noticed that the value for CXCL-10 had a very large standard deviation and so gave a p value of 0.206. Do the authors have an explanation for this? Was it due to one or more patients with particularly severe symptoms?
5. The presentation of the results in Tables 2 and 3 is awkward because the cells in the tables are split onto two pages. It would be better to change the layout so that a table is shown in its complete form on one page.
6. Table 2 is not clear and the title is confusing since it says, “Baseline urinary cytokine levels between controls and patients with recurrent urinary tract infection.” The meaning of Patient Number # is not obvious. There is no heading for Δ as mentioned in the legend to the table. Does the total number of patients in the rUTI group include both those with an initial urinary white blood cell (WBC) count of <10 (n = 10) and those with a urinary white blood cell (WBC) count of >10 (n = 30? Are the values for the recurrent UTI group (n = 40) the means and standard deviations of all of the rUTI patients? The cytokine values for the control group are the same as those in Table 1. Were any of the patients in this control group tested again after one week, one month, or three months? The p values are >2.0 for all of the proteins except for NGAL but some are designated with a * to indicate that they are <0.05. This is not clear. Which values were used to calculate the p values shown in the table? Again, there are very large values for many of the standard deviations.
7. If the goal of the authors in presenting the results in Table 2 is to show which cytokine values decrease over a period of 3 months, it would probably be better to report the data as a series of standard line plots or box plots in which the cytokine concentrations (as means with standard deviations) are plotted as a function of time. Letters could be used to indicate the statistically significant differences from the baseline values. This might indicate better that the concentrations of NGAL, IL-8, and TNF-α go down. The other values show large standard deviations and are not significantly different, so maybe only those three should be included. The rest of the results could be stated in the text.
8. Likewise, I find Table 3 confusing. The main result as noted in the text is that the NGAL concentrations in those patients with a UTI recurrence within one month were higher at the baseline and did not decrease significantly. The concentrations for the patients with no recurrence started off at somewhat high level but then gradually decreased. The baseline values shown for these patients were lower than that shown in Table 1 or Table 2. Why? The values for CXCL-1, IL-8, CXCL-10, MCP-1, and TNF-α were higher in those patients with a UTI recurrence. These values also decreased in the patients with no UTI recurrence but only some of the differences were significant. The use of p values compared to baseline does not help much and so only those with * should be included. Again, it might be better to present the data as a series of standard line plots or box plots in which the cytokine concentrations (as means with standard deviations) are plotted as a function of time. Letters could be used to indicate statistically significant differences from the baseline value.
9. Table 4 is very similar to Table 3 but deals with patients who had a recurrence of a UTI by 3 months. Again, the main result seems to be that high persistent levels of NGAL correlate with a subsequent UTI. It is not clear to me that there are any major differences between the results in Table 3 and those in Table 4, so only one set of data probably needs to be included.
10. As noted in the Discussion, there are a variety of anatomical and physiological that can contribute to a UTI and its recurrence. Did the authors identify any of these in the patients who shows high NGAL concentration?
Author Response
Dear Editor and reviewers:
Thank you for your comments. We have made revision according to your suggestions. The followings are the point-to-point for the individual comment.
Reviewer #1
This manuscript describes the urinary levels of several pro-inflammatory proteins in control patients and those with a history of recurrent urinary tract infections. The goal was to determine if any of these proteins could be used as a predictive biomarker for the diagnosis of recurrent UTIs in other patients. The manuscript is one of a long series of papers from this research group, which have been published in various journals and cited in PubMed. The work appears to have been carefully done and the results are potentially very important. However, there are some missing experimental details and the presentation of the results is awkward. I think that the manuscript could be accepted for publication in the International Journal of Molecular Sciences after major revisions.
The following are more specific comments and suggestions.
- The participants are described as 40 women with a history of recurrent uncomplicated urinary tract infections and 7 age-matched controls. What was the cause of the UTI in those patients that had one? While it might have been Escherichia coli, there are other bacteria that can cause similar infections. The authors say that broad spectrum antibiotics were prescribed as part of an earlier self treatment or the initial clinical assessment, but which ones were used is not given. More extensive microbiological and clinical history would be helpful. As the authors note in the Discussion, this study utilized a small sample size, particularly for the controls. Why were so few patients without symptomatic UTIs used?
Reply: Thank you for the comment. In this study, women with uncomplicated, symptomatic, and urine culture-proven bacterial cystitis were enrolled to investigate whether some urinary cytokines can be used to predict the persistence or recurrence of UTI within 3 months after the initial episode of cystitis, so that antibiotics may be necessary to continue for a complete eradication of UTI. (Lines 68-70)
We have added the urine culture results in Table 1, and the bacterial growth in patients with UTI recurrence within 3 months. (Line 87)
Patients who had UTI persistence or recurrence within 1 month were continuously treated according to previous culture results without urine culture. However, for patients who had UTI recurrence in 1 to 3 months, urine culture was routinely performed. (Lines 279-282) Only one patient had bacteria growth changing from Klebsiella pneumonia to E. coli. The other patients showed the same bacterial growth in the repeat urine culture within 3 months. (Lines 84-86) Because all women had history of rUTI, some patients had self-treatment with antibiotics according to their previous urine culture results and sensitivity test before the visit. (Lines 262-263) The antibiotics including broad spectrum antibiotics such as quinolones or cephalosporines were prescribed. (Lines 265) The controls were women with bladder irritation symptom but negative urinalysis and culture result and without UTI symptoms were also invited to provide mid-stream urine samples. (Lines 267-269)
We did not invite too many controls for the study due to limitation of the study budget.
- The laboratory procedures are described very briefly and only include a listing of the commercial kits used. Reference 14 is given as a source of the methods for analysis, but this reference is actually a review article. I looked online and found an earlier paper from this group which gives these methods in more detail (Jiang, YH et al., 2020, Am. J. Physiol. Renal Physiol. 318: F1391-1399). It would be better either to summarize the methods as was done in this paper or to cite it as a reference. Protein concentrations are given in ng/ml in the legend to Table 1, but pg/ml in the earlier paper by Jiang et al. The authors should include the correct units in the Materials and Methods section.
Reply: Thank you for the comment. We have changed the reference 17, accordingly. (Line 308) We apologized that the correct units of protein concentration should be pg/ml. We have revised the unit in the footnote of Table 2. (Line 108)
- Of the 40 patients with a history of recurrent UTI, 12 had a recurrence in one month and 7 more had a recurrence in one to three months. The total percentage of recurrence thus appears to have been 19/40 or 47.5%. Again, this indicates a small sample size for the analysis. In the case of those patients who showed a UTI recurrence, was there any microbiological analysis? Was the infection due to the same bacterial pathogen as the initial infection or a different one?
Reply: Thank you for the comment. In fact, the UTI within the first month might be persistence of UTI rather than recurrence, we did not routinely culture the urine.
Regarding the 7 women who had UTI recurrence in one to three months, the urine culture showed similar bacteria as the initial urine culture. We have added this statement in the result section. (Lines 85-87)
- Table 1 shows the baseline urinary cytokine levels in the control group (n = 7) and in the recurrent UTI group, subdivided into those with a urinary white blood cell (WBC) count of <10 (n = 10) and those with a urinary white blood cell (WBC) count of >10 (n = 30). The pvalue refers to the difference between the two white blood cell groups. The values for several of the cytokines were significantly higher in the high WBC group than in the low WBC group. Did the authors assign a value to the severity of the clinical symptoms in these patients? Did this correlate with the WBC count? I noticed that the value for CXCL-10 had a very large standard deviation and so gave a p value of 0.206. Do the authors have an explanation for this? Was it due to one or more patients with particularly severe symptoms?
Reply: Thank you for the comment. We also noted that the standard deviations of some urine cytokine levels such as CXCL1, IL-8, and CXCL10 are very large. However, we did not use validated questionnaire to measure the severity of bladder symptoms. The large standard deviation is likely caused by one or more patients with particularly higher grade of bladder inflammation. This also reflect the fact that the case number of this study was small. We have added this in the limitation of the study. (Lines 239- 244)
- The presentation of the results in Tables 2 and 3 is awkward because the cells in the tables are split onto two pages. It would be better to change the layout so that a table is shown in its complete form on one page.
Reply: Thank you for the comments. The layout of tables has been revised to show the whole cells of the table within the same page. However, the editorial office might change the layout which is sending for review.
- Table 2 is not clear and the title is confusing since it says, “Baseline urinary cytokine levels between controls and patients with recurrent urinary tract infection.” The meaning of Patient Number # is not obvious. There is no heading for Δ as mentioned in the legend to the table. Does the total number of patients in the rUTI group include both those with an initial urinary white blood cell (WBC) count of <10 (n = 10) and those with a urinary white blood cell (WBC) count of >10 (n = 30)? Are the values for the recurrent UTI group (n = 40) the means and standard deviations of all of the rUTI patients? The cytokine values for the control group are the same as those in Table 1. Were any of the patients in this control group tested again after one week, one month, or three months? The pvalues are >2.0 for all of the proteins except for NGAL but some are designated with a * to indicate that they are <0.05. This is not clear. Which values were used to calculate the p values shown in the table? Again, there are very large values for many of the standard deviations.
Reply: Thank you for the comment. In this table, the 40 patients with rUTI include both WBC <10/HPF and WBC >10/HPF. The title has been revised and a new figure 1 was added to present the urinary cytokine levels between controls and patients with recurrent urinary tract infection at baseline and different time points after treatment.” (Line 109-118) The controls did not have their urine cytokines tested again at follow-up time points. Because some patients with rUTI did not provide urine at their follow-up visit, that data of urinary cytokines were analyzed according to the true patient number at each follow-up time point.
- If the goal of the authors in presenting the results in Table 2 is to show which cytokine values decrease over a period of 3 months, it would probably be better to report the data as a series of standard line plots or box plots in which the cytokine concentrations (as means with standard deviations) are plotted as a function of time. Letters could be used to indicate the statistically significant differences from the baseline values. This might indicate better that the concentrations of NGAL, IL-8, and TNF-α go down. The other values show large standard deviations and are not significantly different, so maybe only those three should be included. The rest of the results could be stated in the text.
Reply: Thank you for the comment. We have revised the presentation in a bar plot (Figure 1) for the urine cytokine levels in the previous Table 3 to show the changes of urine cytokine levels with time in the patients with rUTI and comparison with the controls. Therefore, the original Table 2 has been deleted. The changes of urine cytokines in patients with rUTI are presented in the text of result section. (Lines 109-118, Figure 1)
- Likewise, I find Table 3 confusing. The main result as noted in the text is that the NGAL concentrations in those patients with a UTI recurrence within one month were higher at the baseline and did not decrease significantly. The concentrations for the patients with no recurrence started off at somewhat high level but then gradually decreased. The baseline values shown for these patients were lower than that shown in Table 1 or Table 2. Why? The values for CXCL-1, IL-8, CXCL-10, MCP-1, and TNF-α were higher in those patients with a UTI recurrence. These values also decreased in the patients with no UTI recurrence but only some of the differences were significant. The use of pvalues compared to baseline does not help much and so only those with * should be included. Again, it might be better to present the data as a series of standard line plots or box plots in which the cytokine concentrations (as means with standard deviations) are plotted as a function of time. Letters could be used to indicate statistically significant differences from the baseline value.
Reply: Thank you for the comment. The urine cytokine levels shown in the original Table 2 were total patients with rUTI, whereas the data shown in the original Table 3 and 4 were from the subgroups of patients with and without UTI recurrence within 1 month (Table 3) or within 1 to 3 months (Table 4). The urine cytokine levels in patients without UTI recurrence (28 within 1 month, and 21 within 3 months) are lower than the total 40 patients because patients with UTI recurrence have higher urine cytokine levels.
According to your suggestion, we have changed the presentation of the urine cytokine levels between patients with and without UTI recurrence within 1 month in a line plots for clear demonstration of the changes of urine cytokine levels at different time points between two subgroups. (Figure 2) The table to demonstrate the cytokine changes with time between total patients with and without UTI recurrence within 3 months is preserved in Table 3.
- Table 4 is very similar to Table 3 but deals with patients who had a recurrence of a UTI by 3 months. Again, the main result seems to be that high persistent levels of NGAL correlate with a subsequent UTI. It is not clear to me that there are any major differences between the results in Table 3 and those in Table 4, so only one set of data probably needs to be included.
Reply: Thank you for the comment. We agree to reduce the number of tables to avoiding confusion. Based on your suggestion, We have shown the changes of urine cytokine levels between patients with and without UTI recurrence within 1 month with time in a line plot in Figure 2. We have deleted the changes of urinary cytokine levels in one month (in original Table 3) and report the significant data in the result section. (Lines 147-154)
- As noted in the Discussion, there are a variety of anatomical and physiological that can contribute to a UTI and its recurrence. Did the authors identify any of these in the patients who shows high NGAL concentration?
Reply: Thank you for the comment. All the enrolled women had been proven uncomplicated and recurrent UTI by videourodynamic study. (Lines 255) We did not find any of them had febrile pyelonephritis or demonstrable functional or anatomical abnormality of the upper urinary tract. (Lines 260-261) Therefore, we did not find the possible renal lesion in the patients with high NGAL concentration.
Reviewer 2 Report
Comments and Suggestions for Authors
Regarding the subjects: The pathogenic microorganisms of cystitis in women change around menopause. Although there is no statistical difference in age between the control group and the group given antibacterial drugs, considering the age, they may have already gone through menopause. This is thought to significantly impact the distribution of bacterial flora in the urinary tract, so this reviewer says it may also affect the research results. In addition, women of this age may have some lower urinary tract symptoms or anatomical abnormalities such as pelvic organ prolapse or urethral stricture (although it has been shown that overactive bladder symptoms were significantly higher in recurrent cases). This is particularly the case in patients with recurrent urinary tract infections. It is known that NGAL levels increase in cases of lower urinary tract dysfunction (neurogenic bladder) and renal dysfunction (AKI and CKD). This reviewer would like to know how thoroughly the patient's lower urinary tract was evaluated (especially from an anatomical perspective) and what the renal function was like. If this point is unclear, I feel that the significance of this study is limited. Please provide detailed comments that will satisfy the reviewer regarding the above points.
As the authors state in the Limitations section, I also feel that including cases that visited the hospital while receiving antimicrobial treatment is a problem. I think that this should be excluded.
In various guidelines for treating urinary tract infections, it is not recommended to use broad-spectrum antibiotics (this study is inappropriate because the details of the kinds of antibiotics are not stated) as a first-line treatment for acute (uncomplicated) cystitis from the perspective of appropriate use of antibiotics. In addition, it is recommended that the treatment period be short, from 3 to 5 days. In this study, broad-spectrum antibiotics were administered for at least one week (this “at least” is puzzling to this reviewer). Furthermore, if symptoms or pyuria persisted, antibiotics were administered for an additional seven days. Treatment of urinary tract infections without considering the bacteriological results is inappropriate. Why is this?
The drug sensitivity of the pathogenic microorganism and the antimicrobial agent used must influence the persistence of inflammation after antimicrobial chemotherapy. Please provide details of the pathogenic microorganism detected, the drug sensitivity test results, the drug sensitivity test, and the results of the drug sensitivity test of the pathogenic microorganism at the time of recurrence. The authors state that long-term antimicrobial administration is desirable in high urinary NGAL markers cases. Still, this reviewer cannot be convinced unless it is assumed that appropriate antimicrobial chemotherapy is being carried out.
As the authors state, recurrent urinary tract infections are determined when two recurrences occur within six months or three recurrences within one year. In the current patient group, the frequency of recurrence within 3 months was high (30% of patients had a recurrence within 1 month), suggesting that treatment based on the patient's background may have been inadequate. The presence or absence of NGAL levels was incidental, and if treatment had been based on the patient's background, would there have been no recurrence? Please comment.
In Table 2, the 40 patients with recurrent UTI include 19 patients with a recurrence during the study period. Still, considering the possibility that each urinary marker may increase again at the time of recurrence, I feel that it is inappropriate to look at the changes over time unless the recurrent cases are excluded at the time of confirmation of recurrence. Similarly, in Table 3, 7 of the 28 patients who did not have a recurrence of UTI within 1 month were included in the seven who had a recurrence of UTI within 3 months. Considering the possibility that the urinary markers may change at the time of recurrence, shouldn't the cases be excluded at the time of recurrence of UTI? The same applies to Table 4. Please state the authors' opinion.
Author Response
Dear Editor and reviewers:
Thank you for your comments. We have made revision according to your suggestions. The followings are the point-to-point for the individual comment.
Reviewer #2
Regarding the subjects: The pathogenic microorganisms of cystitis in women change around menopause. Although there is no statistical difference in age between the control group and the group given antibacterial drugs, considering the age, they may have already gone through menopause. This is thought to significantly impact the distribution of bacterial flora in the urinary tract, so this reviewer says it may also affect the research results. In addition, women of this age may have some lower urinary tract symptoms or anatomical abnormalities such as pelvic organ prolapse or urethral stricture (although it has been shown that overactive bladder symptoms were significantly higher in recurrent cases). This is particularly the case in patients with recurrent urinary tract infections. It is known that NGAL levels increase in cases of lower urinary tract dysfunction (neurogenic bladder) and renal dysfunction (AKI and CKD). This reviewer would like to know how thoroughly the patient's lower urinary tract was evaluated (especially from an anatomical perspective) and what the renal function was like. If this point is unclear, I feel that the significance of this study is limited. Please provide detailed comments that will satisfy the reviewer regarding the above points.
Reply: Thank you for the comment. All the enrolled women had been proven uncomplicated and recurrent UTI by videourodynamic study. (Lines 255) Patients with bladder outlet obstruction, dysfunctional voiding or detrusor underactivity that may risk the UTI recurrence have been excluded from the study. In addition, we did not include any of the patients who had febrile pyelonephritis or demonstrable functional or anatomical abnormality of the upper urinary tract. (Lines 260-261) Therefore, we did not find the possible renal lesion in the patients with high NGAL concentration.
As the authors state in the Limitations section, I also feel that including cases that visited the hospital while receiving antimicrobial treatment is a problem. I think that this should be excluded.
Reply: Thank you for the comment. As we mentioned in the methods section, all these patients had history of recurrent UTI in recent 6 months or 1 year, although some of them took antimicrobials according to their previous urine culture results and sensitivity test before visiting, (Lines 262-263) they still symptomatic and had been diagnosed to have acute cystitis at other clinic. The urine WBC count was still high in part of the patients and urine cytokines were not specifically low. Therefore, it is not inappropriate to include them for follow up the UTI recurrence.
In various guidelines for treating urinary tract infections, it is not recommended to use broad-spectrum antibiotics (this study is inappropriate because the details of the kinds of antibiotics are not stated) as a first-line treatment for acute (uncomplicated) cystitis from the perspective of appropriate use of antibiotics. In addition, it is recommended that the treatment period be short, from 3 to 5 days. In this study, broad-spectrum antibiotics were administered for at least one week (this “at least” is puzzling to this reviewer). Furthermore, if symptoms or pyuria persisted, antibiotics were administered for an additional seven days. Treatment of urinary tract infections without considering the bacteriological results is inappropriate. Why is this?
Reply: Thank you for the comment. As we mentioned in the methods section, all these patients had history of recurrent UTI in recent 6 months or 1 year, although some of them took antimicrobials according to their previous urine culture results and sensitivity test before visiting, they still symptomatic and had been diagnosed to have acute cystitis at other clinic. (Lines 262-263) Urine culture was routinely performed at the first visit and quinolones or cephalosporines were prescribed based on their previous urine culture. We believe this treatment strategy is appropriate. As have added Table 1 to show the baseline urine culture results between patients with and without UTI recurrence.
The drug sensitivity of the pathogenic microorganism and the antimicrobial agent used must influence the persistence of inflammation after antimicrobial chemotherapy. Please provide details of the pathogenic microorganism detected, the drug sensitivity test results, the drug sensitivity test, and the results of the drug sensitivity test of the pathogenic microorganism at the time of recurrence. The authors state that long-term antimicrobial administration is desirable in high urinary NGAL markers cases. Still, this reviewer cannot be convinced unless it is assumed that appropriate antimicrobial chemotherapy is being carried out.
Reply: Thank you for the comment. Patients who had UTI persistence or recurrence within 1 month were continuously treated according to previous culture results without urine culture. However, for patients who had UTI recurrence in 1 to 3 months, urine culture was routinely performed. (Lines 279-282) Only one patient had bacteria growth changing from Klebsiella pneumonia to E. coli. The other patients showed the same bacterial growth in the repeat urine culture within 3 months. (Lines 84-86)
As the authors state, recurrent urinary tract infections are determined when two recurrences occur within six months or three recurrences within one year. In the current patient group, the frequency of recurrence within 3 months was high (30% of patients had a recurrence within 1 month), suggesting that treatment based on the patient's background may have been inadequate. The presence or absence of NGAL levels was incidental, and if treatment had been based on the patient's background, would there have been no recurrence? Please comment.
Reply: Thank you for the comment. In fact, the UTI occurred within 3 months might be persistent or recurrent. Some patients had UTI symptom relapsed after initial antibiotics treatment, while the others might have persistent pyuria despite antibiotics according to urine culture results. This study aimed to investigate potential urinary inflammatory biomarkers and specific urinary inflammatory biomarkers for predicting persistence or recurrence of UTI after the initial antibiotic treatment. Patients with frequently recurrent UTI might have chronic bladder inflammation, resulting in urothelial dysfunction and barrier deficit. This study also show that the urine cytokines declined slowly with time in patients with recurrent UTI, indicating the presence of inflammatory process within the bladder wall, which might render the breakthrough infection of the newly invaded bacteria or intracellular bacteria community. WE have added this statement in the Discussion section. (Lines 224-233)
In Table 2, the 40 patients with recurrent UTI include 19 patients with a recurrence during the study period. Still, considering the possibility that each urinary marker may increase again at the time of recurrence, I feel that it is inappropriate to look at the changes over time unless the recurrent cases are excluded at the time of confirmation of recurrence. Similarly, in Table 3, 7 of the 28 patients who did not have a recurrence of UTI within 1 month were included in the seven who had a recurrence of UTI within 3 months. Considering the possibility that the urinary markers may change at the time of recurrence, shouldn't the cases be excluded at the time of recurrence of UTI? The same applies to Table 4. Please state the authors' opinion.
Reply: Thank you for the comment. We agree the reviewer’s comment. Urine cytokines increase rapidly at the onset of UTI, and will also decline rapidly after antibiotics treatment. As shown in Table 2. Patients who had UTI episode but urinalysis showed WBC ≦10/HPF had significantly lower urine cytokine ;levels compared with those with WBC >10, indicating urine cytokines are sensitive but fluctuate rapidly with UTI episodes and after treatment. Monitoring urine cytokine levels might help us understand the bladder inflammation status and providing low dose antimicrobial prophylaxis for a longer period to adequately eradicate intracellular bacterial communities and improve urothelial barrier function. (Lines 235-238) We have also changed the presentation of urine cytokines levels in patients with UTI recurrence within 1 month to a line plot. The data are quite similar with those in Table 3 which show the urine cytokine levels within 3 months. Patients with UTI recurrence within 1 month also had some elevated urine cytokine levels at 3 months, indicating the presence of unresolved chronic inflammation. Based on these consideration, we do not think putting all the data of these patients in Table 3 is inappropriate. I hope the reviewer can agree with our opinion.
Reviewer 3 Report
Comments and Suggestions for Authors
The authors have performed a study about the use of urinary neutrophil gelatinase-associated 2 lipocalin as a biomarker in recurrent UTI.
1. The authors have not acknowledged the type of study they conducted and thus no EQUATOR Statement was used in the format of this paper.
2. The authors have not assessed a priori the number of patients they included in this study and thus, no validity calculation has been performed. The sample size of this study is limited and the presentation of the findings are confusing.
3.Neutrophil gelatinase-associated 2 lipocalin has been shown to assist on the recognition of renal injury. Perhaps, the elevation of neutrophil gelatinase-associated 2 lipocalin concentrations are due to the renal injury due to serious UTI. No microorganisms are mentioned. Did the recurrent cases of UTI experience cystitis or pyelonephritis?
4. What type of antibiotic did the patients receive? Did they receive any at all? Many drugs have serious adverse effects, such as trimethoprim, for which has been found that in the 14 days after antibiotic initiation for a UTI, trimethoprim was associated with the highest odds of acute kidney injury.
Author Response
Dear Editor and reviewers:
Thank you for your comments. We have made revision according to your suggestions. The followings are the point-to-point for the individual comment.
Reviewer #3
The authors have performed a study about the use of urinary neutrophil gelatinase-associated 2 lipocalin as a biomarker in recurrent UTI.
- The authors have not acknowledged the type of study they conducted and thus no EQUATOR Statement was used in the format of this paper.
Reply: Thank you for the comment. This is an observational study to search for potential urinary inflammatory biomarkers and specific urinary inflammatory biomarkers for predicting persistence or recurrence of UTI after the initial antibiotic treatment in women. (Lines 252-254)
- The authors have not assessed a priori the number of patients they included in this study and thus, no validity calculation has been performed. The sample size of this study is limited and the presentation of the findings are confusing.
Reply: Thank you for the comments. We agree that the did not calculate the sample size for the patients included in this study is a limitation of this study. The small number of control patients could reduce the strength of this study. (Lines 2397-244) However, with the small sample size, the significant findings of persistent elevated urinary NGAL levels found in patients with UTI recurrence within 3 months provide a valuable evidence for clinical practice. (Lines 323-325)
- Neutrophil gelatinase-associated 2 lipocalin has been shown to assist on the recognition of renal injury. Perhaps, the elevation of neutrophil gelatinase-associated 2 lipocalin concentrations are due to the renal injury due to serious UTI. No microorganisms are mentioned. Did the recurrent cases of UTI experience cystitis or pyelonephritis?
Reply: Thank you for the comment. All the enrolled women had been proven uncomplicated and recurrent UTI by videourodynamic study. (Line 255) We did not find any of them had febrile pyelonephritis or demonstrable functional or anatomical abnormality of the upper urinary tract. (Lines 260-261) Therefore, we did not find the possible renal lesion in the patients with high NGAL concentration. In addition, We have added the urine culture results in Table 1. (Line 83-87)
- What type of antibiotic did the patients receive? Did they receive any at all? Many drugs have serious adverse effects, such as trimethoprim, for which has been found that in the 14 days after antibiotic initiation for a UTI, trimethoprim was associated with the highest odds of acute kidney injury.
Reply: Thank you for the comment. The antibiotics including broad spectrum antibiotics such as quinolones or cephalosporines were prescribed. (Lines 265)
Round 2
Reviewer 1 Report
Comments and Suggestions for Authors
The authors have tried to respond to all of my earlier comments about this manuscript and I appreciate their efforts, particularly with respect to the presentation of their data. I have a few additional suggestions at this point.
1. Because the authors are looking at the recurrence or persistence of a urinary tract infection in patients with a history of recurrent UTI (what they label rUTI), it is important to distinguish between the patient’s history and a new recurrence during the three-month period of this study. I would insert the phrase “with a history of recurrent UTI” instead of rUTI wherever there might be ambiguity about this. For example, in lines 72, 73, 81, 89, 97, etc.
2. The names of microorganisms are normally given in italics in published scientific papers. This should be done for Escherichia coli (E. coli) and Klebsiella pneumoniae in line 85 and for all of the bacteria listed in the new Table 1.
3. The text in lines 89-101 after Table 1 needs to be clarified since the data differ from those shown in Table 2. I think the authors mean that patients with a history of recurrent UTI have different baseline levels of IL-8 and NGAL than the uninfected controls. These patients (n = 40) were subsequently followed and 19 had a recurrence of a UTI and 21 did not. The value for IL-8 in the whole rUTI group (99.2 + 135) seems to be the average of both the low WBC group (WBC<10/HPF, n =10) and the high WBC group (WBC>10/HPF, n = 30). The same is true for the NGAL value (319 + 416). While the results for NGAL did not show a statistically significant difference at the p = 0.05 level (p = 0.077), the numbers suggest that there might be an important difference. The authors might explain that this higher p value was due to the large standard deviation.
4. The use of HPF (high power field) as a unit is explained in the Materials and Methods section, but it would be helpful to put into this text or into the legend to Table 2.
5. The authors state that “the levels of the other urinary biomarkers did not differ significantly between the controls and patients with rUTI.” The p values in Table 2 would indicate that they do (except for CXCL10). This is because they refer to a comparison of the WBC<10/HPF group and the WBC>10/HPF group. It might be clearer to calculate the combined average value for both groups and compare that to the value for the control group. The authors could then calculate a p value for this comparison and present it along with the p value comparing the two WBC groups.
6. The text makes a further distinction for NGAL between patients with a WBC of 10-75/HPF and a WBC >100/HPF. Was this done for IL-8 or the other cytokines?
7. The new Figure 1 is a much clearer way to present some of the results and I thank the authors for doing this. Are the units for NGAL ng/ml or pg/ml like the others? It looks like the authors only included the upper lines of the standard errors in this figure, although the text says they calculated a mean + standard error for each compound at each point in time. I think that this was just done for clarity, but it should be stated in the figure legend. The authors use the phrase “overall patients with recurrent urinary tract infection” in the legend to Figure 1. I think they mean all of the 40 patients in this study from both the low WBC group (WBC<10/HPF, n = 10) and the high WBC group (WBC>10/HPF, n - 30). Again, using the phrase a history of recurrent UTI may be appropriate. The baseline bar in the IL-8 graph appears to have a mean of about 220 pg/ml, which differs from the 99.2 pg/ml value mentioned on line 90. I may have missed something here but it should be clarified.
8. The new Figure 2 is good. The text on line 127 says “within 1 month after the initial treatment” but the figure now includes the data out to 3 months. It would be helpful to put the text from lines 147-149 here to indicate that a total of 19 patients had a recurrence of a UTI and 21 did not. Again, for most of the points only the upper error bar is shown. The authors should consistently include both error bars or only one of them.
9. The data shown in Table 3 appear to be same values as those in the new Figure 2. The p values in Table 3 refer to the differences between the Recurrent (n = 19) and No Recurrent (n = 21) groups. I would add this to the legend and then put in *p<0.05 compared to baseline in the same patient group. While some of the p values are < 0.05, many are not. I think that the headings UTI recurrence (n = 19) and No UTI recurrence (n = 21) as in Table 1 are better. I am not sure this table needs to be included. The authors should justify this.
Author Response
Dear Editor and reviewers:
Thank you for your comments. We have made revision according to your suggestions. The followings are the point-to-point for the individual comment.
Reviewer #1
The authors have tried to respond to all of my earlier comments about this manuscript and I appreciate their efforts, particularly with respect to the presentation of their data. I have a few additional suggestions at this point.
- Because the authors are looking at the recurrence or persistence of a urinary tract infection in patients with a history of recurrent UTI (what they label rUTI), it is important to distinguish between the patient’s history and a new recurrence during the three-month period of this study. I would insert the phrase “with a history of recurrent UTI” instead of rUTI wherever there might be ambiguity about this. For example, in lines 72, 73, 81, 89, 97, etc.
Reply: Thank you for the comment. We have deleted the abbreviation of recurrent UTI (rUTI) and replaced with “a history of recurrent UTI” to describe the total 40 patients enrolled in this study. For patients with new episode UTI and had “persistent UTI”, “recurrent UTI” or “no UTI recurrence” after antibiotics treatment, these terms are used throughout the text to avoiding ambiguity.
- The names of microorganisms are normally given in italics in published scientific papers. This should be done for Escherichia coli ( coli) and Klebsiella pneumonia in line 85 and for all of the bacteria listed in the new Table 1.
Reply: Thank you for the comment. The names of microorganisms have been changed to italics as shown in Table 1 and text (Line 91).
- The text in lines 89-101 after Table 1 needs to be clarified since the data differ from those shown in Table 2. I think the authors mean that patients with a history of recurrent UTI have different baseline levels of IL-8 and NGAL than the uninfected controls. These patients (n = 40) were subsequently followed and 19 had a recurrence of a UTI and 21 did not. The value for IL-8 in the whole rUTI group (99.2 + 135) seems to be the average of both the low WBC group (WBC<10/HPF, n =10) and the high WBC group (WBC>10/HPF, n = 30). The same is true for the NGAL value (319 + 416). While the results for NGAL did not show a statistically significant difference at the p = 0.05 level (p = 0.077), the numbers suggest that there might be an important difference. The authors might explain that this higher p value was due to the large standard deviation.
Reply: Thank you for the comment. In Table 2, we have inserted the urine biomarker levels of the 40 patients with a history of recurrent UTI. The p value in Table 2 indicates the statistical analysis of patient with low and high WBC count at baseline urinalysis. (Lines 111-112 in footnotes) For further clarification, we also analyzed the urine biomarkers in patients with WBC count ≤ 10, 10-75, and ≥ 75/HPF. The data are shown in supplemental Table 1. (Lines 102-108)
- The use of HPF (high power field) as a unit is explained in the Materials and Methods section, but it would be helpful to put into this text or into the legend to Table 2.
Reply: Thank you for the comment. We have inserted the “high power field” ahead of the abbreviation of HPF. (Line 100)
- The authors state that “the levels of the other urinary biomarkers did not differ significantly between the controls and patients with rUTI.” The p values in Table 2 would indicate that they do (except for CXCL10). This is because they refer to a comparison of the WBC <10/HPF group and the WBC >10/HPF group. It might be clearer to calculate the combined average value for both groups and compare that to the value for the control group. The authors could then calculate a p value for this comparison and present it along with the p value comparing the two WBC groups.
Reply: Thank you for the comment. The p values in Table 2 is the comparison between patients with a history of recurrent urinary tract infection with baseline WBC ≤ 10/HPF and WBC >10/HPF. (Lines 111-112 in footnotes) We have also inserted the urine levels of biomarkers of the 40 patients with a history of recurrent UTI for clarification.
- The text makes a further distinction for NGAL between patients with a WBC of 10-75/HPF and a WBC >100/HPF. Was this done for IL-8 or the other cytokines?
Reply: Thank you for the comment. The analysis was performed in all urinary biomarkers. The data are shown in supplemental Table 1. (Lines 102-108)
- The new Figure 1 is a much clearer way to present some of the results and I thank the authors for doing this. Are the units for NGAL ng/ml or pg/ml like the others? It looks like the authors only included the upper lines of the standard errors in this figure, although the text says they calculated a mean + standard error for each compound at each point in time. I think that this was just done for clarity, but it should be stated in the figure legend. The authors use the phrase “overall patients with recurrent urinary tract infection” in the legend to Figure 1. I think they mean all of the 40 patients in this study from both the low WBC group (WBC<10/HPF, n = 10) and the high WBC group (WBC>10/HPF, n = 30). Again, using the phrase a history of recurrent UTI may be appropriate. The baseline bar in the IL-8 graph appears to have a mean of about 220 pg/ml, which differs from the 99.2 pg/ml value mentioned on line 90. I may have missed something here but it should be clarified.
Reply: Thank you for the comment. The unit of NGAL is ng/ml and the other biomarkers are pg/ml. We have added the units of biomarkers in the footnotes of the tables. We have added the lower limit of standard error into the bar graph. The statement in footnote has been revised to “in the 40 patients with a history of recurrent urinary tract infection” (Lines 127)
- The new Figure 2 is good. The text on line 127 says “within 1 month after the initial treatment” but the figure now includes the data out to 3 months. It would be helpful to put the text from lines 147-149 here to indicate that a total of 19 patients had a recurrence of a UTI and 21 did not. Again, for most of the points only the upper error bar is shown. The authors should consistently include both error bars or only one of them.
Reply: Thank you for the comment. The new figure 2 demonstrates the changes of urine biomarkers from baseline to 3 months between patients with UTI persistence or recurrence and no UTI recurrence at 1 month after the antibiotics treatment. We have described this in the text (Lines 131-139) and also in the figure legend (Lines 151-153). The SE plus and minus bars have been added in the figure 2. The data are also shown in the supplemental Table 3.
- The data shown in Table 3 appear to be same values as those in the new Figure 2. The p values in Table 3 refer to the differences between the Recurrent (n = 19) and No Recurrent (n = 21) groups. I would add this to the legend and then put in *p<0.05 compared to baseline in the same patient group. While some of the p values are < 0.05, many are not. I think that the headings UTI recurrence (n = 19) and No UTI recurrence (n = 21) as in Table 1 are better. I am not sure this table needs to be included. The authors should justify this.
Reply: Thank you for the comment. Table 3 shows the changes of urine cytokine levels between total patients who had UTI persistence or recurrence and those without UTI within 3 months after the antibiotics treatment. (Lies 156-158) Among the total 40 patients with a history of recurrent UTI, 9 had persistent UTI (at 1M and 3M) and 10 had recurrent UTI (3 at 1M and 7 at 3M) after the latest UTI episode. We have added the analysis among patients with persistent UTI, recurrent UTI, and no UTI recurrence. (Lines 172-180) A new Figure 3 is added and the data are attached in the supplemental Table 4. (Lines 182-187)

Reviewer 2 Report
Comments and Suggestions for Authors
In Table 1, “baseline” refers to the urine culture test results taken at the research institution when the patient was enrolled in this study, right? This is not the result of a test performed before enrollment. If so, I request that you exclude cases where the bacteria could not be identified in culture. This is because it cannot be strictly said to be a recurrent acute bacterial cystitis (you named the title“recurrent”). The description “persistent urinary tract infection” is acceptable. If you must include it, please explain why so everyone can understand. Also, it is usual to italicize the description of bacteria.
Regarding lines 235-238. As long as the antimicrobial treatment to acute uncomplicated bacterial cystitis is appropriate according to the sensitivity test, the "bacteria" should disappear completely after one week unless the same bacteria cause re-infection. On the other hand, as we can see from this study, "inflammation" remains in some patients if suitable treatment is introduced. I guess such a host has a high reactivity to infection, so inflammation persists. Therefore, from the perspective of appropriate use of antimicrobial agents, rather than simply using low-dose antimicrobial agents for long periods, wouldn't it be better to add anti-inflammatory treatment or administer drugs that can regenerate epithelial barrier mechanisms continuously? It is well known that some quinolone antibiotics have anti-inflammatory effects by suppressing cytokines, but can the same be said of cephalosporin and penicillin antibiotics? What kind of drugs are the authors assuming will be administered?
The standard error is shown in Figures 1 and 2, but are the urinary cytokine levels for each case based on the average of multiple measurements, etc.? If you are showing single measurement values, I think it would be better to show the standard deviation, as I would like to see the actual variation in each sample. Please show me why you used the standard error. For Figure 1, I recommend selecting a box plot confirming the individual values for the seven control and 40 UTI cases.
Author Response
Dear Editor and reviewers:
Thank you for your comments. We have made revision according to your suggestions. The followings are the point-to-point for the individual comment.
Reviewer #2
In Table 1, “baseline” refers to the urine culture test results taken at the research institution when the patient was enrolled in this study, right? This is not the result of a test performed before enrollment. If so, I request that you exclude cases where the bacteria could not be identified in culture. This is because it cannot be strictly said to be a recurrent acute bacterial cystitis (you named the title “recurrent”).
The description “persistent urinary tract infection” is acceptable. If you must include it, please explain why so everyone can understand.
Also, it is usual to italicize the description of bacteria.
Reply: Thank you for the comment. In fact, the patients enrolled in this study included 9 patients with persistent UTI and 10 with recurrent UTI after the baseline antibiotics treatment. Within 1 month after the initial treatment, 12 (30%) of 40 patients had UTI persistence (n=9) or recurrence (n=3), and 7 (25%) of 28 patients who did not exhibit UTI persistence or recurrence within 1 month had UTI recurrence within 1–3 months. (Lines131-133)
The urine cultures were performed at the time of visit or several days before the baseline visit. Therefore, the result of urine culture can be considered as the bacterial causing this infection. Patients who had taken antibiotics because they had recurrent UTI and had some antibiotics prepared for the new UTI episodes. Therefore, 10 patients had urinary WBC count ≦10/HPF at baseline urinalysis.
We have deleted the abbreviation of recurrent UTI (rUTI) and replaced with “a history of recurrent UTI” to describe the total 40 patients enrolled in this study. For patients with new episode UTI and had “persistent UTI”, “recurrent UTI” or “no UTI recurrence” after antibiotics treatment, these terms are used throughout the text to avoiding ambiguity.
The names of bacterial have been changed to italics.
Regarding lines 235-238. As long as the antimicrobial treatment to acute uncomplicated bacterial cystitis is appropriate according to the sensitivity test, the "bacteria" should disappear completely after one week unless the same bacteria cause re-infection. On the other hand, as we can see from this study, "inflammation" remains in some patients if suitable treatment is introduced. I guess such a host has a high reactivity to infection, so inflammation persists. Therefore, from the perspective of appropriate use of antimicrobial agents, rather than simply using low-dose antimicrobial agents for long periods, wouldn't it be better to add anti-inflammatory treatment or administer drugs that can regenerate epithelial barrier mechanisms continuously? It is well known that some quinolone antibiotics have anti-inflammatory effects by suppressing cytokines, but can the same be said of cephalosporin and penicillin antibiotics? What kind of drugs are the authors assuming will be administered?
Reply: Thank you for the comment. We agree that it be better to add anti-inflammatory treatment or administer drugs that can regenerate epithelial barrier mechanisms continuously. In this study, we did not use any anti-inflammatory drug to avoiding interference with the bladder inflammation and causing bias to the results of the study. Regarding the antibiotics, we used the antibiotics according to patients’ previous urine culture results, including flouroquinolones such as ciproxin, ofloxacin, and cefmetazole, etc. (Lines 307-309)
The standard error is shown in Figures 1 and 2, but are the urinary cytokine levels for each case based on the average of multiple measurements, etc.? If you are showing single measurement values, I think it would be better to show the standard deviation, as I would like to see the actual variation in each sample. Please show me why you used the standard error. For Figure 1, I recommend selecting a box plot confirming the individual values for the seven control and 40 UTI cases.
Reply: Thank you for the comment. Figure 1 was made to include only NGAL, IL-8, NGF, and TNF-α according to the suggestion from reviewer #1. However, because the standard deviations of each cytokine are large, therefore, using SEM would be better to show the levels of each cytokine in this figure. Due to large standard deviation and changes at different time points after antibiotics treatment, a box plot for the changes of cytokine levels did not provide a clear demonstration.

Reviewer 3 Report
Comments and Suggestions for Authors
The authors have tried to assess the association of various organokines to the recurrence of UTIs, as biomarkers.
1. The Introduction does not mention the rationale. It just mentions the various studies along with speculations. The elevated organokines may be due to the irritation of the inflammation of the tissue involved.
2. Moreover various references do not correctly refer to the fact, the authors mention. E.g. 12, 13.
3. Table 3 demonstrates that NGAL shows significant statistical differences only in NGAL concentrations.
4. How do the authors assess the connection? What statistical analysis have they performed, apart from the repeated measures ANOVA? Which by the way, it is not mentioned anywhere in the paper.
5. Again, the authors did not use any known EQUATOR statement and their results are not reproducible.
6. The sample is too small for the strong association, they mention.
Author Response
Dear Editor and reviewers:
Thank you for your comments. We have made revision according to your suggestions. The followings are the point-to-point for the individual comment.
Reviewer #3
The authors have tried to assess the association of various organokines to the recurrence of UTIs, as biomarkers.
- The Introduction does not mention the rationale. It just mentions the various studies along with speculations. The elevated organokines may be due to the irritation of the inflammation of the tissue involved.
Reply: Thank you for the comment. This study aimed to investigate potential urinary inflammatory biomarkers and specific urinary inflammatory biomarkers for predicting persistence or recurrence of UTI after the initial antibiotic treatment in women so that antibiotics treatment may be necessary to continue for a complete eradication of UTI. (Lines 70-73)
- Moreover various references do not correctly refer to the fact, the authors mention. E.g. 12, 13.
Reply: Thank you for the comment. We have revised the statements in the Introduction for the cited references. (Lines 66-70)
- Table 3 demonstrates that NGAL shows significant statistical differences only in NGAL concentrations.
Reply: Thank you for the comment. The NGAL level in patients with recurrent or persistent UTI show significantly higher than those with no UTI recurrence at baseline and different time points after antibiotics treatment, suggesting NGAL is a sensitive biomarker for UTI persistence or recurrence. The other biomarker CXCL-1 and IL-8 show a less reduction at 3 months in recurrent UTI group, suggesting the patients might have a higher bladder inflammation. (Lines 268-271)
- How do the authors assess the connection? What statistical analysis have they performed, apart from the repeated measures ANOVA? Which by the way, it is not mentioned anywhere in the paper.
Reply: Thank you for the comment. We admit the connection between urinary biomarker levels and bladder inflammation and recurrent UTI are not very solid. This is a preliminary and observational study. (Lines 294-296) We hypothesized the connection based on the changes of urinary biomarkers among patients with persistent UTI, recurrent UTI, and no UTI recurrence compared for the declination with time. For a solid evidence, we need to initiate a prospective study to test the sensitivity and positive predictive value for the urinary NGAL level in patients with and without UTI recurrence.
- Again, the authors did not use any known EQUATOR statement and their results are not reproducible.
Reply: Thank you for the comment. The STROBE was used for checklist of the study and manuscript preparation. (Supplement 5)
- The sample is too small for the strong association, they mention.
Reply: Thank you for the comment. We admit that the enrolled case number of this preliminary study was small. However, with the small patient number, the results of study still provide a significant clinical implication of the important role of urinary NGAL level to predict the persistence or recurrence of UTI in patients with a history of recurrent UTI. (Lines 283-286)

Round 3
Reviewer 3 Report
Comments and Suggestions for Authors
No further comments.